# Domestic Generative Acts and Life Satisfaction among Supplementary Grandparent Caregivers in Urban China: Mediated by Social Support and Moderated by Hukou Status

**DOI:** 10.3390/ijerph182211788

**Published:** 2021-11-10

**Authors:** Haoyi Guo, Steven Sek-yum Ngai

**Affiliations:** Department of Social Work, The Chinese University of Hong Kong, Hong Kong, China; syngai@cuhk.edu.hk

**Keywords:** grandparent caregiver, domestic generative acts, life satisfaction, urban China, Hukou, informal and formal social support

## Abstract

Urban China is witnessing a growth of migrant grandparents apart from the prevalent local grandparent caregiving. However, the health consequences and influencing factors of grandparent caregiving remain largely unknown among migrant and local grandparent caregivers. This study examined informal and formal social support’s mediation roles between domestic generative acts and life satisfaction, as well as investigating Hukou’s (household registration system) moderation effect. Our sample compromised 1013 grandparent caregivers (Migrant = 508, Local = 505) from 12 kindergartens with a multistage clustered random sampling from Eastern China. Migrant grandparent caregivers had significant lower informal social support (M = 4.000, L = 4.355, *p* < 0.001), formal social support (M = 1.787, L = 2.111, *p* < 0.001), and life satisfaction (M = 3.323, L = 3.574, *p* < 0.001) than local ones. Structural equation modeling results indicated that domestic generative acts positively associated with life satisfaction (*b* = 0.085, *p* < 0.05), informal (*b* = 0.223, *p* < 0.001) and formal social support (*b* = 0.080, *p* < 0.05); informal (*b* = 0.379, *p* < 0.001) and formal social support (*b* = 0.138, *p* < 0.001) positively associated with life satisfaction. In addition, both informal (β = 0.084, CI [0.039, 0.101], *p* < 0.001) and formal social support (β = 0.011, CI [0.001, 0.018], *p* < 0.05) mediated the relationship between domestic generative acts and life satisfaction. Furthermore, Hukou status moderated the indirect path from domestic generative acts to life satisfaction via informal social support (*p* < 0.01), but not formal social support (*p* > 0.05). Migrant grandparent caregivers, with limited formal social support resources, were found to be more dependent on informal social support than locals. The findings revealed social support and wellbeing disparities among migrant and local grandparent caregivers in urban China. Theoretical contributions and practical implications are also discussed.

## 1. Introduction

With greater longevity and lower fertility, grandparent caregiving is increasingly available in China and worldwide, and its influencing factors on grandparent caregivers’ health have drawn considerable scholarly attention [1,2,3]. A basic typology to describe grandparent caregiving is suggested by Baker and Silverstein [4]: primary care for grandchildren due to family crisis or parental out-migration and supplementary care to assist working parents. The distinction between the two scenarios is the presence of parents [5]. Previous studies in China mainly focused on rural areas, where grandparents served as primary caregivers for left-behind children under parental out-migration [6,7,8]. On the other hand, supplementary grandparent caregiving in urban China is understudied. It is noteworthy that the background of grandparent caregivers in urban China is increasingly heterogeneous, as many rural grandparents join migrant families in urban destinations to provide childcare [9]. The latest statistics from the National Health Commission of the People’s Republic of China estimate approximately eight million grandparent caregivers migrating to cities [10]. Under China’s Hukou (household registration) system, migrants without an urban residency status have limited access to formal welfare that only locals are entitled to and may experience inequalities in mental health [11]. Moreover, late-life migration may engender language barriers and adaptation disadvantages in building social networks and social contact [12].

While informal and formal social support have generally been beneficial to primary grandparent caregivers [13,14], it remains uncertain how they will affect supplementary grandparent caregivers in China, especially those who have migrated. Though current research has discovered the physical and mental health rewards of grandparent caregiving compared with noncaregivers, it neglects to measure the content and strength of grandparent caregiving with a valid scale by merely using the time spent in grandchild care [15]. In addition, using secondary data that do not target grandparent caregivers may miss essential variables and lack theoretical depth [16]. As a form of generative acts and expression of generativity [17], grandparent caregiving in the urban China context needs further investigation to address these knowledge gaps. Thus, the purpose of this study is to examine how supplementary grandparent caregiving affects life satisfaction and uncover the influencing mechanisms of informal/formal social support. Further, considering the welfare division and social support gap brought by Hukou status, we aim to explore the differences between migrants and locals in the mediated relationships via social support.

### 1.1. Domestic Generative Acts and Life Satisfaction

According to Erikson [18], generativity originally refers to the middle adulthood’s developmental task of establishing and contributing to the next generation; the commitment to younger generations continues into older age as an unprecedented increase in life expectancy elongates the generativity vs. stagnation stage [19,20]. As Aubin and McAdams [21,22] proposed, such generative commitment would actually lead to generative acts in various ways, including offering, nurturing and mentoring. For older adults, generative acts may include caring for grandchildren inside the family and mentoring young people outside the family [23]. In this study, we are interested in the grandparent caregivers’ generative acts inside the family as grandparent caregivers, noted as domestic generative acts that differ from generative acts in the community [24]. In the Chinese context, grandparent caregivers may help out in doing housework when caregiving grandchildren, both of which are considered as domestic generative acts that facilitate the middle generation in the labor market, especially for mothers [25]. Adopting generativity theories may contribute to understanding grandparent caregiving from a life course perspective and connecting to grandparent caregivers’ wellbeing [20]. Western research has acknowledged being generative as an essential factor in maintaining late-life psychological health because it may provide a sense of purpose, feeling valued, and feelings of productivity [26,27]. A sample of 300 Spanish supplementary grandparents reported personal growth when they conducted generative acts to grandchildren’s socialization [28]. In the Chinese context, grandparent caregiving as a common practice of being generative in old age may enhance the psychological wellbeing of grandparent caregivers, as revealed from a Hong Kong study [29]. Based on the above theories and findings, it is rational to assume a positive association between domestic generative acts and life satisfaction of supplementary grandparent caregivers in China.

### 1.2. Informal and Formal Social Support as Mediators

Social support is defined as the perception or experience that an individual is cared for and responded to by people in their social groups [30]. Distinguished by its providers, social support can be classified as informal support from family, friends, or neighbors and formal support delivered by professionals [31]. According to the Resourcefulness Theory [32], individuals have the capabilities to develop help-seeking strategies to acquire social resourcefulness in achieving personal goals. By conducting domestic generative acts, grandparents may be involved in social relations in the family and community, such as exchanging childcare ideas with peers or consulting professionals about child nutrition knowledge. Older adults assuming caregiving roles have been found more likely to develop social ties within their network than noncaregivers; in being generative, grandparent caregivers may reply on social support resources around them for the sake of younger generations [33]. Thus, we expect that a higher level of domestic generative acts would predict higher levels of informal/formal social support.

Further, Social Convoy Theory suggests that family members, friends, and other personal networks are crucial social relationship sources and vital elements of one’s wellbeing [34]. Social support relations varying in quantity and quality have significant health implications, especially in later life [35]. When carrying out domestic generative acts, grandparent caregivers may connect with peer caregivers as a source of informal social support or health professionals as a source of formal social support, both of which may impact their wellbeing. We then assume that higher informal and formal social support levels are related to higher life satisfaction among supplementary grandparent caregivers. Moreover, the Convoy Theory recognizes that better quality social ties are more significantly associated with health outcomes; thus, we expect different strengths of informal and formal social support effects on life satisfaction. In addition, evidence from rural China presents that emotional support partially mediates between caregiving and grandparents’ self-reported health [36]. Taken together, we hypothesize that informal and formal social support could be mediators linking domestic generative acts and life satisfaction among supplementary grandparent caregivers in urban China.

### 1.3. Hukou Status as a Moderator

The Hukou system is a nationwide household registration system that assigns each citizen a rural or urban Hukou type based on birthplace and lineage [37]. In the 1950s, the Chinese government used Hukou to control internal migration and maintain rural–urban divisions in welfare provision [38]. Although the control over migration relaxed after 1978 reforms, formal state welfare is still greatly attached to Hukou status; whether a local resident or not may imply disparate entitlement to welfare resources, such as medical insurance [39]. Compared to local counterparts, migrant grandparent caregivers have less access to formal welfare support, leaving informal networks such as family members and friends of greater importance for them. Moreover, as observed by Neo-familism [40], contemporary China is witnessing a decline in the traditional authority of the older generation when grandparents migrate to care for the third generation. Migration at an older age makes grandparent caregivers more dependent on the middle generation economically and emotionally [41]. In other words, the degree of informal and formal social support influences may differ between migrant and local grandparent caregivers. We thus propose Hukou status as a moderator in the relationships from domestic generative acts to life satisfaction via informal and formal social support.

### 1.4. The Present Study

As reviewed above, few studies have rigorously measured domestic generative acts of supplementary grandparent caregivers in urban China and investigated the effect on life satisfaction, and the mediating mechanism of social support remains unknown. Moreover, despite the emergence of grandparents in migration and the accompanying social welfare and support gap, little attention has been paid to the moderating role of Hukou status on social support’s influencing mechanism. To fill the research gap, we formulated the following hypotheses:

**Hypothesis** **1 (H1).**
*Domestic generative acts positively predict life satisfaction among supplementary grandparent caregivers.*


**Hypothesis** **2 (H2).**
*Domestic generative acts increase supplementary grandparent caregivers’ informal/formal social support levels, which, in turn, predict a higher level of life satisfaction.*


**Hypothesis** **3 (H3).**
*Hukou status moderates the mediation effects of informal/formal social support between domestic generative acts and life satisfaction.*


## 2. Materials and Methods

### 2.1. Participants and Procedure

A total of 1013 participants were recruited from 12 kindergartens in Cixi City of Eastern China from September 2020 to November 2020. The sample size was sufficient according to the sample size calculation formula for a cross-sectional survey: $n=\frac{Z^{2}P(1-P)}{e^{2}}$. The study adopted a multistage cluster random sampling. First, four districts were randomly selected in Cixi City, then three kindergartens in each district were randomly selected. In each kindergarten, grandparents caring for children of working parents (only one grandparent of each child) constituted the research sample, including both migrant and local Hukou holders. This study was approved by the Survey and Behavioral Research Ethics Committee at the institution where the authors were affiliated. Informed consent was obtained before the survey. Table 1 presents an overview of the demographic characteristics of the research sample categorized by Hukou status. Grandmothers accounted for 67.7% and 75.8% of the migrant (N = 508) and local (N = 505) samples, respectively. The average age was 56.5 years (SD = 8.7) among migrant grandparent caregivers, four years younger than the local group. The dominant majority had junior or below education level (94.6% for migrant, 88.5% for local). Less than half (44.7%) migrant grandparent caregivers had local social security, while the proportion doubled (87.7%) among local ones. Only 9.5% of migrant grandparent caregivers had access to local medical insurance, yet it was nearly full coverage (95.8%) for the locals.

### 2.2. Measurement

#### 2.2.1. Domestic Generative Acts

Domestic generative acts were measured by the four-item domestic generative acts scale developed by Cheung [29] from a sample of Hong Kong older adults and validated with a Chinese sample of grandparents [24]. Participants were asked to indicate the frequency of doing domestic generative acts on a five-point Likert scale ranging from “1 = Almost none” to “5 = Very often.” The four-item statements include a dimension of caregiving activities encompassing two items: “I take care of children and grandchildren when they are ill” and “I take care of the grandchildren when their parents are not available,” as well as a dimension of instrumental activities encompassing two items: “I take care of my offspring’s daily life, including preparing meals” and “I do housework for my children.” Each item was used as an indicator of the latent construct of domestic generative acts. A higher sum score indicated a higher level of domestic generative acts. Cronbach’ s alpha for this scale was 0.897 in the present study.

#### 2.2.2. Social Support

Informal social support was assessed in the 12-item Multidimensional Scale of Perceived Social Support (MSPSS), consisting of three four-item subscales of family, friends, and significant others (MSPSS, [42]). Participants responded to the statements describing their informal social support using a five-point Likert scale ranging from “1 = Strongly disagree” to “5 = Strongly agree.” Example items would be, “There is a special person who is around when I am in need” and “My friends really try to help me.” The mean scores of each subscale constituted the latent construct of informal social support. A higher sum score suggested a higher level of informal social support. The scale showed satisfactory internal reliability in the present sample (Cronbach’ s α = 0.928).

Formal social support was assessed by our self-developed four-item scale based on its definition and the context of China and with reference to a similar study among Chinese rural elders [43]. Participants were asked the frequency they used four categories of formal social support: (1) community activities, such as elderly recreational activities organized by the neighborhood committee, (2) state subsidies, such as medical allowance, (3) NGO services, such as educational workshops for grandparents, and (4) professional consultation, such as health information inquiry. The answers were responded by utilizing a five-point Likert scale ranging from 1 = “Very rarely” to 5 = “Very often.” All items were indicators of the latent construct of formal social support. A higher sum score indicated a higher formal social support level. Cronbach’s alpha for this scale was 0.920 in the present study.

#### 2.2.3. Life Satisfaction

Life satisfaction was measured using the five-item Satisfaction With Life Scale (SWLS, [44]) on a five-point Likert scale ranging from “1 = Strongly disagree” to “5 = Strongly agree.” Participants responded to statements such as “In most ways, my life is close to my ideal” and “I am satisfied with my life” based on their feelings. All items composed the latent construct of life satisfaction. A higher sum score suggested higher life satisfaction. Cronbach’s alpha for this scale was 0.906 in the present study.

#### 2.2.4. Hukou Status

The Hukou system in mainland China divides citizens into rural and urban Hukou (household registration) types. Though some rural migrants can acquire an urban Hukou, Hukou conversion is complex and rare [45]. If a person’s household registration is not the regular resident place, they are considered a migrant instead of a local Hukou holder. In the present study, Hukou status was a dichotomous variable measured by whether a grandparent caregiver hosts a local Hukou or not: 1 = local, 0 = migrant.

#### 2.2.5. Covariates

The following covariates were controlled in the analysis: gender (1 = Male, 2 = Female); age; education level (1 = Primary school or below, 2 = Junior school, 3 = Senior school, 4 = University or college or above); marital status (1 = Married, 2 = Divorced, 3 = Widowed); monthly income (1 = 0–2500 RMB, 2 = 2501–5000 RMB, 3 = 5001 RMB or above); whether or not having chronic diseases (1 = Yes, 0 = No); whether or not having local medical insurance (1 = Yes, 0 = No); whether or not having local social security (1 = Yes, 0 = No).

### 2.3. Data Analyses

First, descriptive statistics were conducted in SPSS 24.0 (released by IBM Corp. in 2016, Armonk, NY, USA) to generate the means, standard deviation of continuous variables, percentage of categorical variables, followed by bivariate correlation analysis among key variables. A two-tailed *p*-value smaller than 0.05 demonstrated statistical significance. Second, an independent sample *t*-test was operated to examine whether the difference in key variables was statistically significant between the migrant and local groups. Next, structural equation modeling was implemented in Mplus 8.0 (released by Muthen & Muthen in 2017, Los Angeles, CA, USA) with a two-step procedure to test the research hypotheses. Confirmatory factor analysis first tested the measurement model to check how observed variables represented the latent variables. Then, the structural model validated influencing paths among key variables, using the maximum likelihood method to examine the mediating roles of informal and formal social support from domestic generative acts to life satisfaction. The following model fit indices were considered to evaluate goodness-of-fit: the Chi-square value (χ^2^), the comparative fit index (CFI), the Tucker-Lewis index (TLI), root mean square error of approximation (RMSEA), and standardized root mean square residual (SRMR). The criteria were above 0.90 on CFI and TLI and below 0.08 on the RMSEA and SRMR [46]. We applied the 2000-sample bootstrapping method with 95% bias-corrected confidence intervals (CI). The confidence’s exclusion of zero meant a statistically significant estimate [47]. After securing a validated structural model with mediation effect, we further used Hayes PROCESS macro (Model 14) to spontaneously test the moderated mediation effects [48]. All covariates were controlled in the analysis process.

## 3. Results

### 3.1. Preliminary Analyses

Descriptive statistics and bivariate correlations for main variables are displayed in Table 2. As expected, domestic generative acts (with the mean scores of each item ranging from 3.45 to 3.98 out of 5.0) were positively correlated with informal social support (*r* = 0.230, *p* < 0.01), formal social support (*r* = 0.077, *p* < 0.05), and life satisfaction (*r* = 0.156, *p* < 0.01). In addition, informal social support (*r* = 0.419, *p* < 0.01) and formal social support (*r* = 0.240, *p* < 0.01) were positively correlated with life satisfaction. Moreover, we examined the differences in domestic generative acts, social support, and life satisfaction between migrant and local grandparent caregivers through an independent-sample *t*-test. The results are shown in Table 3. Compared with local counterparts, migrant grandparent caregivers reported lower levels of informal social support (*t* = 6.919, *p* < 0.001), formal social support (*t* = 5.011, *p* < 0.001), and life satisfaction (*t* = 4.399, *p* < 0.001).

### 3.2. Testing for the Mediation Effect

A structural equation model was adopted to examine the mediation effects of informal and formal social support between domestic generative acts and life satisfaction. We first used confirmatory factor analysis to ensure that all latent variables of domestic generative acts, informal social support, formal social support, and life satisfaction were well-represented by observed indicators. The measurement model showed satisfactory model fit: χ^2^ (98) = 256.007, *p* < 0.001, CFI = 0.985, TLI = 0.982, RMSEA = 0.040, and SRMR = 0.032. We further required all factor loadings to exceed 0.40, based on the guideline proposed by Brown [49]. The results of the measurement model are presented in Table 4.

Subsequently, the structural model also demonstrated good model fit (see Figure 1): χ^2^ (195) = 505.533, *p* < 0.001, CFI = 0.973, TLI = 0.964, RMSEA = 0.040, and SRMR = 0.045. The results showed that domestic generative acts had a direct effect on life satisfaction (*b* = 0.085, SE = 0.035, *p* < 0.05), which supported Hypothesis 1. Moreover, domestic generative acts positively predicted grandparent caregivers’ informal social support (*b* = 0.223, SE = 0.039, *p* < 0.001) and formal social support (*b* = 0.080, SE = 0.03, *p* <0.05); informal social support (*b* = 0.379, SE = 0.039, *p* < 0.001) and formal social support (*b* = 0.138, SE = 0.033, *p* < 0.001), in turn, positively predicted grandparent caregivers’ life satisfaction. A bootstrap procedure of 2000 samples was generated from the original dataset (N = 1013) via random sampling. The indirect effects of domestic generative acts on life satisfaction were 0.084 via informal social support (SE = 0.019, 95% CI [0.039, 0.101], *p* < 0.001) and 0.011 via formal social support (SE = 0.005, 95% CI [0.001, 0.018], *p* < 0.05). The 95% biased-corrected confidence interval did not contain zero, which verified the indirect relationships between domestic generative acts and life satisfaction via informal and formal social support. Thus Hypothesis 2 gained full support.

### 3.3. Testing for Moderated Mediation

It was expected that Hukou status would moderate the second stage of the mediation process. As shown in Table 5, in model 3, the effect of Hukou status was significant (*b* = 0.661, SE = 0.289, *p* < 0.05). To be specific, the interaction term of informal social support*Hukou status was significant (*b* = −0.177, SE = 0.065, *p* < 0.01), while formal social support*Hukou status was nonsignificant (*p* > 0.05). The index of moderated mediation of informal social support (difference between conditional indirect effects) was −0.032 (95% CI= [−0.059, −0.007]). There was a significantly stronger indirect relation between domestic generative acts and life satisfaction via informal social support among migrant grandparent caregivers (*b* = 0.085, SE = 0.016, 95% CI= [0.056, 0.118]) than that of local ones (*b* = 0.053, SE = 0.012, 95% CI = [0.032, 0.079]). However, there was no such significant difference in indirect effect via formal social support (*p* > 0.05). Moreover, the effect of informal social support on life satisfaction was significantly stronger among migrant grandparent caregivers (*b* = 0.475, SE = 0.045, *p* < 0.001) than that of local ones (*b* = 0.298, SE = 0.048, *p* < 0.001), but such difference was not significant for formal social support (*p* > 0.05). Thus Hypothesis 3 is partially supported. For the descriptive purpose, we plotted the relationship between informal social support and life satisfaction, separately for migrant and local groups (see Figure 2).

## 4. Discussion

The present study has tested the moderated mediation of Hukou status on the effects of domestic generative acts on life satisfaction via informal and formal social support in an integrated model. The findings suggested a significant association between domestic generative acts and life satisfaction, mediating effects of informal and formal social support, and Hukou status’s moderating effect on informal social support. Major research hypotheses were generally tested via first-hand empirical data.

As revealed by the preliminary analysis, local grandparent caregivers have a double proportion in chronic disease (Local = 35.6%; Migrant = 17.7%) than that of migrant group, substantiating the Healthy Migrant Theory [50]. Despite that, a significantly higher level of life satisfaction was documented among locals than migrants (Local = 3.574; Migrant = 3.323), in line with the Hukou-associated mental health disadvantages among older migrants in China [38]. A similar significant disparity was found in informal social support (Local = 4.355; Migrant = 4.000) and formal social support (Local = 2.111; Migrant = 1.787), suggesting that local Hukou grandparent caregivers tended to have more fruitful supporting resources than the migrants, which supported previous literature concerning the negative effect of migration on social capital among Chinese older adults [12].

The study confirmed the first hypothesis by showing that supplementary grandparent caregivers in urban China with more domestic generative acts were rewarded greater life satisfaction, which echoed the generativity theory [51]. The findings were also aligned with foreign literature regarding the positive outcomes of grandparents’ generativity, claiming that grandparent caregiving on a part-time basis helped grandparents’ personal growth and meaning-making [27,28]. In addition, grandparent-provided childcare, as a way of facilitating the maternal labor force participation in urban China, may boost the economic wellbeing of the extended family and add to the self-worth of grandparent caregivers, thus enhancing their life satisfaction [25]. The particular cultural expectation ascribed to grandparent caregiving that grandchildren would show more filial piety to grandparents who cared for them may also contribute to grandparents’ life satisfaction, as revealed from evidence in rural China [52].

Moreover, by conducting domestic generative acts, grandparent caregivers would create more interactions within families and in the communities [53]. For example, caregiving a grandchild has been recognized to positively associated with an individual’s network size and support in German community-dwelling adults [54]. On the other hand, when higher levels of support from friends, friends, and others are perceived, grandparent caregivers might feel greater life satisfaction, and lower informal and formal social support may predict lower life satisfaction. According to the Social Convoy Model [34], the size and quality of social relationships might engender different health consequences. The fact that migrant grandparent caregivers had lower informal and formal social support levels may further deepen the wellbeing gap [55]. Taking the two paths together, the mediating mechanism of informal and formal social support (Hypothesis 2) was fully established. This finding corresponded with the Western literature about informal social support as a mediator between functional status and quality of life among older adults [56]. Moreover, it was in a similar situation with custodial grandparents where informal social support mediated the association between depression and mental health [57], and formal social support contributed to grandparents’ resilience and wellbeing [58].

More importantly, the study uncovered that the strength of social support’s indirect effect differs in migrant and local grandparent caregivers by examining the moderating effect of Hukou status in Hypothesis 3. The indirect effect from domestic generative acts via informal social support was more substantial in the migrant group. Although having a significantly lower level of informal social support, migrant grandparent caregivers were substantially more dependent on informal social support from personal networks than locals to gain life satisfaction. To our speculation, migrant grandparent caregivers with little formal state welfare protection regarding medical insurance and social security in host cities may resort to informal support instead. In the current sample, only one in ten migrant grandparent caregivers has been equipped with medical insurance in the host place, and less than half possess social securities. The life satisfaction of grandparent caregivers with a migration background would be more heavily affected by their informal resources, such as family members or friends, given the deficiency in formal services due to the Hukou barrier [59]. While for the local grandparent caregivers, the support from informal networks exhibited fewer influences on life satisfaction as other formal welfare provisions would have spared the part. Of note, the two groups did not differ in the mediation effect of formals social support. One possible explanation could be the relatively low level of formal social support for grandparents in China. Both migrant (mean:1.787) and local groups (mean: 2.111) scored less than 3.0 out of 5.0 on the formal social support measure of a Likert scale, indicating the frequency of using formal support services was lower than “Sometimes.”

Several limitations are acknowledged in the study. Firstly, the cross-sectional nature of the research design limits its causal inference about the relationships in the model. Thus, future research may consider a longitudinal study exploring causal directions, the long-term effect of domestic generative acts on life satisfaction, and the effect of migration duration on life satisfaction. Secondly, migrant grandparent caregivers are not registered officially, so it is difficult to strictly randomize the targeting group. We used kindergartens as the sampling frame, and only grandparents caring for children at 3 to 6 years old (the age range of children in kindergarten) were included. Additional research could be conducted about grandparents caring for infant babies (0–3 years old) to consolidate the findings. In addition, the measure of formal social support was self-developed and had not been tested among various populations in the Chinese context before, and the data collection site was located in Eastern China, which may lack representativeness for other parts of China. Despite these limitations, this study contributes to the literature about social support and life satisfaction among supplementary grandparent caregivers in urban China.

This study has several important theoretical and practical implications. Theoretically, the moderated mediation model proposed by the study provides a possible framework for future studies to explore the direct and indirect relationships between domestic generative acts and life satisfaction among Chinese grandparent caregivers. While previous studies mainly studied grandparent caregiving in rural settings [6,8] or cities without migration [60], this study captures the effect of migration among grandparent caregivers, with the special consideration of Hukou influence. Although the positive influence of informal social support on grandparents’ health has been discussed, its mediating role between domestic generative acts and life satisfaction had rarely been tested until we made the first attempt. Moreover, the present study extends the literature by indicating that the indirect path from generative acts to life satisfaction via informal social support is moderated by Hukou status, that migrant grandparent caregivers are affected to a greater extent by informal social support levels than locals, given their scarce formal welfare support due to residency restriction. From this perspective, we provide robust empirical evidence to represent the disparate social support situations (informal and formal) and consequently the distinctive life satisfaction of migrant and local grandparent caregivers in urban China.

Regarding the practical implications, this study reflects the imperative for Hukou reform and adjustment of welfare provision in China at the policy level, considering the welfare division and following health disparities among migrants and locals. Though an employment-based social insurance system has made state welfare available for working migrants [59], it still excludes migrant grandparent caregivers who are usually not employed while caring for grandchildren in urban destinations. The selectivity of welfare provision has inevitably increased the health risks of the migrant group of grandparent caregivers, which threatens their wellbeing. Thus, this study calls for a citizenship-based welfare provision rather than a Hukou-based one to address the health needs of migrant grandparents and the migrant population at large [61]. Moreover, since migrant grandparent caregivers join the migrant families in caregiving grandchildren, chances of international conflicts may increase regarding childcare views [62]. The stronger effect of informal social support among migrant grandparent caregivers implies a more vital impairment on life satisfaction when such informal support fails, with limited formal support to supplement. Interventions with migrant families or group work with migrant grandparent caregivers are needed to improve their informal social support level from families and local friends, thus creating protective networks for grandparent caregivers.

## 5. Conclusions

Our study has filled the knowledge gaps by examining supplementary grandparent caregiving’s positive impact on life satisfaction, revealing the mediating mechanism of informal and formal social support, and demonstrating Hukou status’ moderation effect on informal social support. Caring for grandchildren as a cultural norm is an expression of generativity in China that contributes to grandparent caregivers’ life satisfaction, consistent with Western literature. Though informal and formal social support both showed a mediation effect, the influence of informal social support was more significant for migrant grandparent caregivers than locals, yet, such difference was not displayed in the formal social support. Given the considerable social support and services gap between migrants and locals, state welfare policy reform and more social support interventions are recommended for the migrant grandparents to benefit their life satisfaction in urban China.

## Figures and Tables

**Figure 1 ijerph-18-11788-f001:**
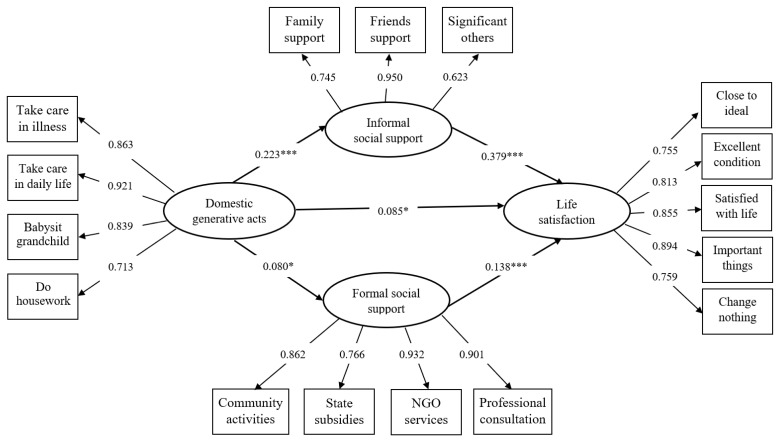
Structural model of domestic generative acts, informal/formal social support, and life satisfaction. χ^2^ (195) = 505.533, *p* < 0.001, CFI = 0.973, TLI = 0.964, RMSEA = 0.040, SRMR = 0.045. * *p* < 0.05; *** *p* < 0.001.

**Figure 2 ijerph-18-11788-f002:**
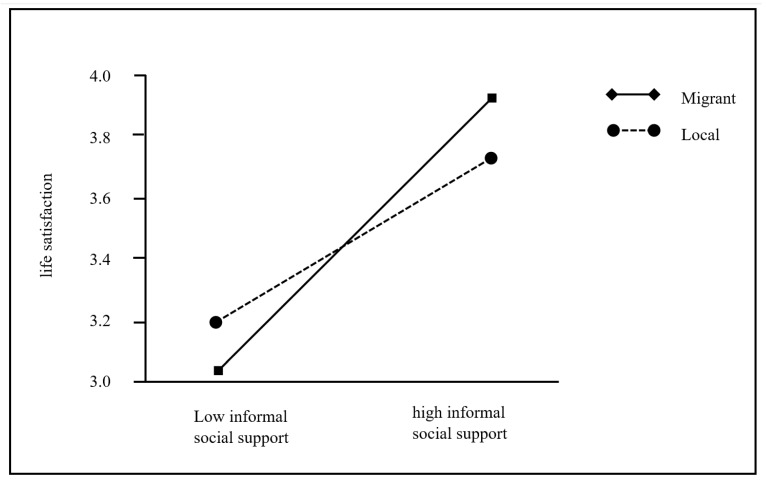
Hukou as a moderator of the relationship between informal social support and life satisfaction.

**Table 1 ijerph-18-11788-t001:** Demographic information of participants (N = 1013).

		Migrant (N = 508)	Local (N = 505)
		%	%
Gender	Male	32.3	24.2
	Female	67.7	75.8
		Mean/SD	Mean/SD
Age (years)	Range: 40–93	56.5/8.7	60.5/5.6
Education	Primary school	67.1	58.0
	Junior school	27.5	30.5
	High school	5.2	9.9
	College or above	0.2	1.6
Marriage	Married	88.4	92.5
	Divorced	2.8	1.0
	Widowed	8.9	6.5
Monthly income	0–2500	62.2	55.0
(RMB)	2501–5000	27.0	35.2
	Above 5001	10.8	9.7
Chronic disease	Yes	17.7	35.6
	No	82.3	64.4
Medical insurance	Yes	9.5	95.8
	No	90.5	4.2
Social security	Yes	44.7	87.7
	No	55.3	12.3

**Table 2 ijerph-18-11788-t002:** Descriptive statistics and bivariate correlations for key variables (N = 1013).

	Range	Mean	S.D.	1	2	3	4
1. Domestic generative acts	1–5	3.675	1.056	1			
2. Informal social support	1–5	4.177	0.835	0.230 **	1		
3. Formal social support	1–5	1.949	1.041	0.077 *	0.248 **	1	
4. Life satisfaction	1–5	3.448	0.915	0.156 **	0.419 **	0.240 **	1

Note: * *p* < 0.05, ** *p* < 0.01 (2-tailed).

**Table 3 ijerph-18-11788-t003:** Differences of main variables by Hukou status.

	Hukou Status	M	S.D.	*t*
Domestic generative acts	Migrant	3.526	0.920	4.544 ***
	Local	3.825	1.157	
Informal social support	Migrant	4.000	0.836	6.919 ***
	Local	4.355	0.797	
Formal social support	Migrant	1.787	0.993	5.011 ***
	Local	2.111	1.063	
Life satisfaction	Migrant	3.323	0.944	4.399 ***
	Local	3.574	0.867	

Note: *** *p* < 0.001.

**Table 4 ijerph-18-11788-t004:** Results of the measurement model.

Latent Variable	Observed Variable	Factor Loading
Domestic generative acts	Take care in illness	0.864
	Take care in daily life	0.921
	Babysit grandchildren	0.838
	Do housework	0.713
Informal social support	Family support	0.749
	Friend support	0.944
	Significant others	0.625
Formal social support	Community activities	0.862
	State subsidies	0.766
	NGO services	0.932
	Professional consultation	0.901
Life satisfaction	Close to ideal	0.756
	Excellent condition	0.813
	Satisfied with life	0.858
	Important things	0.896
	Change nothing	0.763

Note: χ^2^ (51) = 154.514, CFI = 0.986, TLI = 0.982, RMSEA = 0.045, and SRMR = 0.028.

**Table 5 ijerph-18-11788-t005:** Results of moderated mediation analysis for life satisfaction.

Variable	Model 1 (Informal Social Support)			Model 2 (Formal Social Support)			Model 3 (Life Satisfaction)		
	*b*	SE	*t*	*b*	SE	*t*	*b*	SE	*t*
Gender	−0.054	0.059	−0.921	−0.228	0.074	−3.084 **	0.100	0.061	1.645
Age	0.005	0.004	1.470	0.015	0.005	3.299 **	0.005	0.004	1.229
Education	−0.008	0.039	−0.209	0.028	0.049	0.559	0.039	0.040	0.969
Marriage	0.051	0.048	1.068	0.006	0.060	0.104	0.007	0.049	0.146
Monthly income	0.069	0.040	1.711	0.110	0.050	2.184 *	0.108	0.041	2.618 **
Chronic disease	0.016	0.058	0.279	0.011	0.073	0.155	−0.025	0.060	−0.420
Local medical care	0.185	0.062	2.977 **	0.071	0.078	0.914	−0.007	0.108	−0.062
Local social security	0.125	0.063	1.999 *	0.369	0.078	4.697 ***	0.123	0.065	1.901
Domestic generative acts	0.179	0.024	7.335 ***	0.087	0.031	2.851 **	0.057	0.026	2.229 *
Informal social support							0.475	0.045	10.603 ***
Formal social support							0.100	0.038	2.663 **
Hukou							0.661	0.289	2.287 *
Informal social support*Hukou							−0.177	0.065	−2.715 **
Formal social support*Hukou							0.030	0.051	0.587
*R* ^2^	0.090			0.080			0.218		
*F*	11.044 ***			9.749 ***			19.921 ***		

Note: * *p* < 0.05; ** *p* < 0.01; *** *p* < 0.001.

## Data Availability

The datasets generated during and/or analyzed during the current study are not publicly available due to datasets containing information that could compromise the privacy of research participants. The data that support the findings of this study are available from the corresponding author (H.G.) upon reasonable request.

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
