# Peer review of "Domestic Generative Acts and Life Satisfaction among Supplementary Grandparent Caregivers in Urban China: Mediated by Social Support and Moderated by Hukou Status"

_ijerph, 2021, doi:10.3390/ijerph182211788_

Round 1
Reviewer 1 Report
The current manuscript examines the mediating effect of formal and informal social support systems on the relation between domestic generative acts and life satisfaction in supplementary caregiving grandparents in China; this relation was further examined using a moderating factor of migration status within China. The review of theory and strong methodology are significant strengths of the reviewed manuscript. The manuscript details findings that could influence both current understanding from a research perspective, as well as policy considerations. However, there are currently a few limitations that, if addressed, I believe will strengthen the manuscript and increase its potential impact.
- Regarding the review of generativity in Erikson’s theory: The generativity vs stagnation stage was originally focused within middle adulthood (ages 40-65). While the current sample includes individuals predominately within this age bracket, the variation with the study goes beyond middle adulthood such that 1+ standard deviation for both local and migrant populations go beyond the age of 65. The authors may choose to highlight research such as Kim and colleagues (2017) which provides a more modern conceptualization of generativity or speak upon how Erikson’s theory was expanded to include older individuals as well.
- Kim, S., Chee, K. H., & Gerhart, O. (2017). Redefining generativity: Through life course and pragmatist lenses. Sociology Compass, 11(11), e12533. doi: 1111/soc4.12533
- Regarding the specific focus on domestic generative acts: Additional detail should be included on what encompasses domestic generative acts. The scale developed by Cheung (2009) appears to reflect actions focused on activities of daily living within the household or general household care. Further justification is needed as to how these are generative acts that benefit, or are done with the intent to benefit, future generations. McAdams and de St. Aubin (1992) described generative acts as being expressed through means of creation, maintenance, or offering. Specific examples of behaviors are provided and may be included by authors to justify their use of this scale to encompass generative action.
- McAdams, D. P., & de St. Aubin, E. (1992). Theory of generativity and its assessment through self-report, behavioral acts, and narrative themes in autobiographies. Journal of Personality and Social Psychology, 62(6), 1003-1015.
- Regarding the covariates section within the methodology: While a reasonable inference can be made regarding why the selected covariates have been included within the study, more explicit justification should be made. Is there research to support why these covariates may influence the relation between study variables?
- Regarding data analyses: Were there any cases of missing data within the data set or were other forms of data cleaning essential? If so, please note these considerations. Were the underlying assumptions for each analysis performed checked prior to data analysis? Additionally, I assume that the sample size met sufficient power requirements for the modeling analyses, but include a statement about this met requirement if it is true.
- Regarding the t-test analyses: Authors should include the effect sizes for each t-test. Statistical significance can only indicate so much and if the study is seeking to influence policy or suggest policy changes, the clinical significance of the differences between local and migrant populations can be better inferred by effect sizes for the analyses.
- Regarding the factor analyses conducted: A few of the factor loadings were about 0.9, which may be worth discussing. When considering the general rules of thumb regarding the sizes of factor loadings, anything above 0.9 may be viewed as loading onto the constructs at too high of a level. The authors should address this to justify the items inclusion in the final constructs.
- Regarding the figure of the SEM model: Include a notation of what the *-included symbols mean. The authors did a good job of noting these meanings in previous tables, so just include it in figures as well so that the meaning is clear as a stand-alone figure.
Thank you for the opportunity to review this interesting and meaningful work. I look forward to its eventual publication.
Reviewer 2 Report
Please see attached.
This is a very strong article which tackles an interesting and important topic with sophisticated methodology and good data. I have only a few suggestions for improving the article.
- The abstract is confusing. It assumes knowledge that a lot of readers will not have. I recommend that “supplementary caregiving in urban areas” and “Hukou” be defined briefly here. The references to Resourcefulness Theory and Social Convoy Theory could be dropped from the abstract. They are important in the background literature, but not in the abstract.
- The conceptualization and measurement of “Generative Acts” is an important contribution of this paper. However, readers need a bit more discussion of this topic. Since all participants in the study are caregivers, and presumably caregiving itself is a generative act, this “floor” should be acknowledged. That opens the door for the possibility of a richer conceptualization of the four-item generative acts scale as a measure of intensity and/or nature of caregiving. The four items seem to be tapping two related but separate dimensions: instrumental generative acts (helping with daily life, doing housework), and caring for grandchildren (when they are ill or when parents are not available). I’m not suggesting additional analyses, but it would strengthen the paper to learn more about the patterns and types of caregiving.
- A related issue is the reporting of scores on the generative acts scale. Since there were four items measured on a 5-point Likert scale, how is the range of scores 1 to 5 as reported in Table 2? Tell us more about the distribution of scores on this central concept. Similarly, more detail about the other measures is necessary for readers to fully understand your analyses. How many items were there on the life satisfaction scale, and on the informal support scale? As with the generative acts, how is it possible that the range reported in Table 2 is only 1 to 5 for these multi-item scales?
Round 2
Reviewer 1 Report
I am satisfied with the edits made by the authors in their latest round of revisions and thank them for their efforts.